# From Iron to Copper: The Effect of Transition Metal Catalysts on the Hydrogen Storage Properties of Nanoconfined LiBH_4_ in a Graphene-Rich N-Doped Matrix

**DOI:** 10.3390/molecules27092921

**Published:** 2022-05-03

**Authors:** Alejandra A. Martínez, Aurelien Gasnier, Fabiana C. Gennari

**Affiliations:** 1Consejo Nacional de Investigaciones Científicas y Técnicas (CONICET) and Centro Atómico Bariloche (CNEA), Avenue Bustillo 9500, San Carlos de Bariloche R8402AGP, Argentina; andreaalejandra.m5@gmail.com (A.A.M.); gennari@cab.cnea.gov.ar (F.C.G.); 2Instituto de Nanociencia y Nanotecnología, Nodo Bariloche, San Carlos de Bariloche R8402AGP, Argentina; 3Instituto Balseiro, Universidad Nacional de Cuyo, San Carlos de Bariloche R8402AGP, Argentina

**Keywords:** nanoconfined LiBH_4_, mesoporous carbon matrix, coordination metal catalyst

## Abstract

Incipient wetness impregnation was employed to decorate two N-doped graphene-rich matrixes with iron, nickel, cobalt, and copper nanoparticles. The N-doped matrix was wetted with methanol solutions of the corresponding nitrates. After agitation and solvent evaporation, reduction at 800 °C over the carbon matrix promoted the formation of nanoparticles. The mass of the metal fraction was limited to 5 wt. % to determine if limited quantities of metallic nanoparticles catalyze the hydrogen capture/release of nanoconfined LiBH_4_. Isotherms of nitrogen adsorption afforded the textural characterization of the matrixes. Electronic microscopy displayed particles of definite size, evenly distributed on the matrixes, as confirmed by X-ray diffraction. The same techniques assessed the impact of LiBH_4_ 50 vol. % impregnation on nanoparticle distribution and size. The hydrogen storage properties of these materials were evaluated by differential scanning calorimetry and two cycles of volumetric studies. X-ray diffraction allowed us to follow the evolution of the material after two cycles of hydrogen capture-release. We discuss if limited quantities of coordination metals can improve the hydrogen storage properties of nanoconfined LiBH_4_, and which critical parameters might restrain the synergies between nanoconfinement and the presence of metal catalysts.

## 1. Introduction

The ever-growing anthropogenic demand for energy is still primarily quenched with fossil fuels. Inevitably, this appetite will unleash two banes: their finitude and the greenhouse effect. Sustainable energies should supersede this dependence, but their alternative nature complicates their implementation into the existing energy matrix. Hydrogen is considered an ideal energy vector, except for its low density [1]. Solid-state metal storage systems based on metal hydride benefit from their high gravimetric potential and improved security but may suffer from their slow hydrogen release/capture kinetics [2]. The rise of nanomaterials proposed 1D, 2D, and 3D materials to bypass these limitations [3,4,5].

Lithium borohydride (LiBH_4_) is a prime example of the potential and limitations of metal hydrides [6]. LiBH_4_ presents high theoretical gravimetric (18 wt. %) and volumetric (121 kg m^−3^) hydrogen density [7,8], but its application suffers from the high temperatures (500 °C) of hydrogen release and high pressure (100 bars) of uptake [6,9,10]. Nanoconfinement is a promising route to improve the kinetic and thermodynamic properties of LiBH_4_ [11,12]. Reducing the particle size of this hydride to the nanoscale improves mass transport rates, and the contact/tension of the matrix also alters phase transformations of LiBH_4_ [13,14,15,16,17,18] and reaction pathways [19,20]. The nanoconfinement of LiBH_4_ lowers the orthorhombic to hexagonal phase transition temperature and greatly enhances species mobility, the hydride behaving as an ionic liquid from transition temperature (circa 120 °C) [21,22,23]. Yet, nanoconfinement alone did not solve all the issues of LiBH_4_ hydrogen storage application: it is still difficult to release all available hydrogen below 300 °C, and reversibility hardly exceeds 60% [6,24,25]. Indeed, the mechanism of pore wetting by LiBH_4_ during impregnation and the expulsion of LiH during hydrogen release are pivotal to the material’s life cycle [26,27]. The own mass of the matrix also quickly limits LiBH_4_ specific interest by lowering the material’s hydrogen mass capacity; for example, a matrix presenting 1.0 cm^3^/g pore-volume filled at 50% suffers a 14 to 3.5 wt. % mass capacity loss. Reducing pore size and pore filling generally lowers the onset of hydrogen release, but unfortunately, it also reduces the available pore volume, thus the mass of LiBH_4_ (i.e., of H_2_) per mass of material. Even worse, the reversibility of hydrogen uptake/release may suffer from this strategy [6,28,29,30]. In a recent study, we demonstrated that the gradual release of hydrogen from nanoconfined LiBH_4_ can be related to the gradual filling of the pores, the smaller being filled first and releasing hydrogen at a lower temperature [31]. The fast exchange rate of the mobile nanoconfined LiBH_4_ is probably responsible for the gradual hydrogen release in a matrix constituted of both micro- and mesopores [21,31].

Thus, even if nanoconfined LiBH_4_ presents much better performance than its bulk counterpart, it might not be enough to attain technological applications. Considering this, the strategies used to improve bulk LiBH_4_ could help to enhance further the properties of nanoconfined LiBH_4_ [6,24,25]. Many illustrated how combining carbon nanomaterials and metallic nanoparticles improved hydrogen release/capture. For example, cobalt-decorated carbon nanotubes presented a synergistic effect over MgH_2_ sorption properties [32], while palladium displayed an improved hydrogen uptake capacity when decorating N-doped graphene [33]. Regarding specifically the nanoconfinement of LiBH_4_ in a carbon matrix, Ngene et al. decorated a porous matrix with nickel nanoparticles [34], Xian et al. doped a carbon network with carbon nanotubes and TiO_2_ nanoparticles [35], and Wang et al. presented N-doped carbon nanospheres [36]. Exceptionally, Chen et al. replaced the carbon matrix by a reactive Ni-based porous materials [37]. However, carbon derivatives remain a major focus, and graphene in particular is a widely studied carbon nanomaterial with outstanding properties and wide potential applications that have attracted great interest [38]. For instance, graphene wrapping promoted a synergistical activation of LiBH_4_ in the presence of catalysts [39,40]. Accordingly, our group proposed to take advantage of this nanomaterial by its sequential inclusion within a carbon matrix [28], its doping with heteroatoms [29], and decoration by coordination metals [30]. The latter advanced materials presented nickel, cobalt, and their mixture, improving hydrogen release and capture, but it was unclear if this was a catalytic or a mass effect. Indeed, we chose to decorate a matrix of relatively wide pores (9 nm) with coordination metals at 5 vol. % (of the mesopore volume, 0.75 cm^3^/g), representing more than 100% of the actual mass of LiBH_4_ in the 70 vol. % LiBH_4_ filled material (almost three times the mass of LiBH_4_ in the 30 vol. % filled material). The disproportion between LiBH_4_ (0.67 g/cm^3^) and coordination metal (around 9 g/cm^3^) densities strongly limits both the technical application of the material (worsening the hydrogen´s mass capacity) and the physicochemical interaction of the species (the metal volume being much lower than that of its hydride counterpart). The ability to disperse smaller metallic particles all over the matrix would improve their interaction with the hydride without compromising the mass capacity of the system. It is crucial to homogeneously decorate the inner parts of the pores (where the *activated* LiBH_4_ is), not only the surface of the matrix. Recent theoretical studies highlight the interest of nickel, cobalt, copper, and iron in catalyzing the dehydrogenation of LiBH_4_ [41], but it is still difficult to simulate the behavior of hundreds to thousands LiBH_4_ units in a 2–5 nm cavity [42]. Hence, it is unclear if the kinetic effect of a metal dopant over H-desorption can translate from the bulk to the nanoconfined space within a matrix activated by a Lewis base [43].

Here, we present the effects of four coordination metals (Fe, Ni, Co, Cu) decorating at 5 wt. % two N-doped matrixes of small pore diameters (4 and 6 nm) filled at 50 vol. % with LiBH_4_. We aimed to determine if a catalytic effect of a coordination metal can enhance the reversible hydrogen release properties of nanoconfined LiBH_4_ in an N-doped matrix. Regarding our previous work, the method proposed here emphasizes reducing the size of the metallic nanoparticles decorating the matrix and improving their homogeneity. We also employed matrixes of smaller pores with different nitrogen doping to determine if synergies enhance both effects. The textural parameters of those materials were determined by isotherms of nitrogen adsorption. The coverage of the matrixes by metallic nanoparticles was estimated by SEM (scanning electron microscopy) with elemental mapping. To evaluate the impact of the material´s life on the crystallographic nature of the nanoparticles, PXRD (powder X-Ray diffraction) was performed before impregnating the matrixes by LiBH_4_, just after impregnation, and after two cycles of hydrogen release/uptake. The influence of the nanoparticles over the release/uptake of hydrogen from nanoconfined LiBH_4_ was assessed with DSC (differential scanning calorimetry) and volumetric experiments.

## 2. Results

### 2.1. Chemical and Morphological Characterizations of the Hydride-Free Matrixes

#### 2.1.1. Textural Characterization of the Matrixes

Non-decorated N-doped matrixes

The hydrogen release/uptake properties of nanoconfined LiBH_4_ are profoundly dependent on the matrix pore-size and its filling value: the smaller the pores and the lower their filling, the lower the temperature needed to release hydrogen [31]. Here, we focused on matrixes of relatively small mesopores (from 4 to 6 nm wide), with a non-negligible proportion of micropores, enriched with graphene and doped with nitrogen [29]. X-ray photoelectron spectroscopy (XPS) of these materials can be found in our previous work. Figure 1 presents the isotherms of nitrogen adsorption of the non-decorated, non-impregnated matrixes. Both matrixes were obtained with distinct concentrations of N-dopant, but employing the same concentrations of resin precursors, affording sharp pore-size distribution according to BJH (inset). Two distinct peaks were obtained depending on the proportion of ethylene–diamine crosslinker in the graphene hydrogel. As previously observed, the material doped with more nitrogen (G2N) presented pores of smaller size (3.8 nm) than the material doped with less nitrogen (6.1 nm, GN), probably because nitrogen acts as a base-catalyst for the reticulation of the precursors [29]. Despite the peaks´ sharpness, a non-negligible proportion of the total pore volume was constituted by micropores (0.16 cm^3^ in both cases, representing 23% of GN total pore volume, up to 43% of G2N pore volume, see Table 1). As discussed before, the micropores are likely to be filled faster than the mesopores at a given filling value, enhancing the nanoconfined behavior of the impregnated LiBH_4_. We increased the proportions of graphene (+50 wt. %) and N-dopant (+25 mol. %) relative to our previous work, but it did not markedly affect the definition of the peaks [29].

N-doped matrixes decorated with nanoparticles

The evolution of the textural parameters when decorating the matrixes is presented in Table 1 and in the Appendix A. Regarding the material with less nitrogen (GN), the limited decrease of the mass-related parameters is attributable to the mass increase due to the addition of 5 wt. % of metal to the matrix (Table 1). Yet, in the case of the material doped with more nitrogen (G2N) and decorated with Fe or Co, an increase of the mesopore volume (+35%) was observed at the expense of the micropore volume (−60%); while pore-size and overall pore volume were mostly unaffected, the specific area of the material (highly dependent on the micropores) declined notably (−40%). Remarkably, the concerned peak at 3.8 nm was always present in our material but usually in limited proportions. Further characterization techniques will step up the specificity of these matrixes (G2N Fe and Co). It is noteworthy that reducing the micropore volume should raise the onset temperature of the material [31].

#### 2.1.2. Characterization of Nanoparticle’s Size and Distribution

In previous studies, we highlighted that the presence of nanoparticles affects the performance of the hydride [30]. Reducing the relative mass of the metallic element would improve the hydrogen capacity of the material, limit particle size, and determine if a catalytic quantity of metal can positively affect the hydride’s degradation temperature. Previously, we observed Ni, Co, and NiCo particles of limited size (22 ± 6, 27 ± 9, 22 ± 6 nm, respectively) but the presence of large metallic domains translated into a larger average particle size according to the Scherrer’s equation (58, 60, 45 nm, respectively). We hoped to reduce the extent of metallic domains by employing incipient wetness impregnation of the saline solution instead of manual milling of the salt. We chose methanol as it is an excellent solvent of nitrates, its low surface tension is more likely to wet the inner pores of the matrix, and its weaker hydrogen bonding makes it easier to evaporate than water or ethanol.

SEM observations of N-doped matrixes decorated with nanoparticles

Figure 2 presents the typical distribution of Fe, Co, Ni, and Cu nanoparticles over GN and G2N matrixes. Elemental mapping confirmed the attribution of the particles to their corresponding element (see the Appendix A, where M = Fe, Co, Ni, Cu). Cu-decorated materials (D, H) present much bigger particles (>100 nm) than other metals (~20 nm); the larger Cu crystals (with well-defined facets) develop upward longitudinally past 100 nm. The manual measure of at least 300 nanoparticles from five or more representative pictures afforded their statistical size, presented in Table 2 and distribution histograms (Appendix A). These data and the careful observation of higher-magnification images (Appendix A) indicate that the materials with more nitrogen (G2N) present bigger particles with broader distribution and a less defined shape. By comparing Figure 2D,H, the latter presented particles of less defined outer shape, suggesting that the nitrogen-metal interaction is higher and restrains the formation of Cu crystals (see also Appendix A). The same phenomenon seemed to occur for smaller particles of Fe, Co, and Ni, even if not as obvious. Contrarily to our previous observations [30], we did not observe areas covered with extensive metallic structures, indicating that reducing the [metal mass]/[matrix surface] ratio and/or the incipient wetness impregnation favors a more homogeneous repartition of the metal over the matrix. Yet, if the presence of wide crystals was prevented here, the average observed particle size remained similar. Elemental analysis by EDS (energy dispersive microscopy) of the surface of our materials at distinct energies (3 KeV, 15 KeV) indicated that Fe, Co, or Ni metals were present at values very close to the expected 5 wt. % (see the corresponding Appendix A for the EDS). Given the penetration of the beam is highly dependent on its energy (50 nm at 3 keV, 1 um at 15 keV), and as similar results were obtained at distinct energies, the repartition of the metal is likely homogeneous through the thickness of the matrix. It suggests that incipient wetness impregnation efficiently distributes those metals within the whole sample, at least up to the penetrating value of the beam of higher energy. On the other hand, for Cu-decorated matrixes, EDS at 3 keV revealed a mass concentration closer to 20 wt. %, while at 15 keV this value dropped to 5 wt. %. As Cu nanoparticles tend to agglomerate to form much larger structures, they are likely less able to remain within a constricted pore.

Structural characterization of N-doped matrixes with and without nanoparticles

The nature and average size of the nanoparticles were determined by powder X-ray diffraction measurements, presented in Figure 3. Except for the matrixes decorated with Cu, broad peaks were observed, a good indication that large crystal domains were absent. The average crystallite obtained by Scherrer’s equation (Table 2) appeared very close to the one obtained from manual counting. It indicates large crystals are absent, in opposition to our previous work [30], depicting the interest of the current method. Only in the case of metallic nickel, the average particle size is a bit higher according to the equation than from counting, but it is still much closer in values than previously. While the diffractograms of Cu-, Co-, and Ni-decorated matrixes displayed the expected metallic peaks, almost every PXRD pattern (except for G2N Cu) also presented a non-negligible fraction of their corresponding oxide. In the case of GN Fe and G2N Fe, only Fe_3_O_4_ (or possibly Fe_2_O_3_) could be observed, despite a shoulder being present close to 42.8°, accounting for possible narrow metallic particles. It was not the case in our previous work for Ni and Co nanoparticles, albeit then the diffractograms were obtained in the air over several hours, revealing the formation of an oxide layer did not occur rapidly at room temperature. In the current work, an air-tight dome was employed for each sample (accounting for the broad peak between 15–30°), and the samples were exposed to the air for only a short time (less than 5 min) between pyrolysis and vacuum activation. Two factors might have promoted the formation of these oxides: the preponderance of smaller, more reactive particles (namely, because less metal was employed) or the use of methanol. To discriminate between both factors, the same amount of nickel nitrate was manually ground with the matrix, submitted to pyrolysis, and vacuum activated. This material was submitted to PXRD, and no trace of oxide was observed, implying methanol was responsible for the presence of oxides (Appendix A). While the methanol was carefully removed by roto-evaporation and the samples were dried overnight under a flux of dry nitrogen, some solvent likely coordinated with the salt or wet the matrix, affecting the reduction of the metal during the pyrolysis. We were surprised to observe such differences with the ground material; indeed, in any case, we employed nitrate hydrates, the matrix quickly moistened due to air humidity, and the oxides typically formed during pyrolysis were supposed to be fully reduced by the carbon matrix under high temperature in an inert flow. Still, to prevent the formation of oxides, the powders could be placed under a high vacuum at 150 °C (below nitrate melting temperature) for several hours to remove any trace of solvent. In any case, two points must be remembered here: (i) the impregnation of the material is realized under high pressures of H_2_, and we ensured the oxides were not present after this step (see Appendix A); (ii) oxides effectively catalyze the liberation of H_2_ from LiBH_4_, so their presence in the decorated material might not necessarily be prejudicial to the final hydrogen storage material [35,39,44,45,46].

Another intriguing point raised by PXRD was the appearance in specific samples of an unexpected peak at 26.0°. Whilst we could not definitively attribute this peak, its position is similar to the peak of graphite [47,48,49]. It appeared only for G2N matrixes and was more intense for Fe and Co (it was absent from Cu and a shoulder was barely observable with Ni). Interestingly, the same materials presented distinct nitrogen adsorption isotherms, indicating that both the structure and texture of these matrixes were affected during the pyrolysis.

### 2.2. Chemical and Morphological Characterizations of the LiBH_4_-Impregnated Matrixes

SEM observations of just-impregnated N-doped matrixes decorated with nanoparticles

Particles on matrixes just impregnated with LiBH_4_ appeared slightly larger (Figure 4), with a broader distribution (Appendix A), denoting an interaction between the metal and the hydride. This interaction was particularly impressive in the case of G2N Cu (Figure 4H), with splatter-like nanoparticles (see elemental attribution in Appendix A). Higher magnification (Appendix A) indicated that this process could occur with other metals, as the shape of their respective nanoparticles was less defined. Unfortunately, elemental mapping does not allow us to efficiently differentiate boron from carbon, especially at this magnification. Still, single point EDS indicated that boron was present on metallic nanoparticles. The increase in particle size might account for an external layer of boron derivative coating the metal. Similar to the non-impregnated samples, bigger and less-defined nanoparticles were observed for higher proportions of N-dopant.

In some cases, a binder appeared between the nanoparticles (Figure 4D and Appendix A) that can be attributed to the nanoparticles melting together during pyrolysis or LiBH_4_ melting and gluing these during its impregnation. It should be stated here that the presence of LiBH_4_ between the nanoparticles might account for an affinity between the metal and the hydride. This being so, this affinity might compete with the insertion of the hydride within the matrix. Molten structures were also observed (Appendix A) and in a few cases, we were able to observe extensive spilling at the surface of some chunks of matrix (Appendix A). Once, flat structures were observed outside of the matrix of G2N50 Fe (Appendix A). EDS suggested that higher proportions of boron were present when these structures were observed (Appendix A). It should be noted that our samples were exposed to air for a very short time (<2 min) when inserting the sample in the SEM vacuum chamber. Elemental analysis indicates hydrolysis of LiBH_4_ occurred, as higher proportions of oxygen were measured with boron. Given LiBH_4_ degradation products occupy a larger volume than the fresh hydride, it might explain how boron derivatives were observed outside of the matrixes even if filled at 50 vol. %. The observation of cracks in the matrix when spilled boron was observed also suggested the same expulsion mechanism, so the just-impregnated sample might not necessarily present notable amounts of LiBH_4_ outside of the matrix, as we will see by DSC.

Some intriguing behavior observed on boron-rich particles, that are far beyond the scope of this article, are presented in the Appendix A for the reader’s curiosity (Appendix A).

Structural characterization of N-doped matrixes with and without nanoparticles

To gain further insight into the impact of LiBH_4_ impregnation on decorated matrixes, we present the PXRD patterns in Figure 5, and Table 3 presents the characteristic peaks of our materials. While LiBH_4_ impregnated 50 vol. % of each matrix, none of its typical crystallographic peaks (2 Ɵ = 17.7°; 23.7°; 24.7°; 25.6°) was present in any of our samples. It is specific to LiBH_4_ impregnated in small pores to lose its crystalline long-range order. Yet, when the nanoparticles were absent, a peak was observed at 12.6°, typical of our N-doped matrixes once impregnated [29]. We already proposed that this peak might appear by removing elements of symmetry of the orthorhombic cell of LiBH_4_. We are not sure if this could be related to the flat structures observed for G2N50 Fe (Appendix A), as this peak was missing when metallic nanoparticles were present, as already observed for Ni and Co [30].

As previously [30], the peak associated with transition elements suffered a loss in intensity once LiBH_4_ impregnated the matrix, confirming an interaction between the hydride and the nanoparticles. In the case of GN, only Cu was observed, while in G2N Co and Ni were still observed. It must be highlighted that (i) oxides were not observed, (ii) Cu particles were less affected by LiBH_4_ impregnation as they were larger, (iii) G2N50 Fe presented two peaks that could be related to Fe_2_B species, (iv) the peak at 26.0° remained mostly unchanged. While metal particles of comparable size (but more dispersed) were observed by SEM after the impregnation with LiBH_4_, the PXRD patterns of these samples indicated the loss of their crystalline structure.

### 2.3. Evaluation of Hydrogen Release and Characterization of Material’s Evolution

#### 2.3.1. Differential Scanning Calorimetry

DSC is a convenient technique for the rapid evaluation of nanoconfinement over hydrogen release. Figure 6 summarizes the impact of N-doping and nanoparticle decoration over the distinctive peaks of nanoconfined LiBH_4_: phase transition, melting, and hydrogen release. Usually, the loss of long-range order causes these peaks to be flattened and shifted toward lower temperatures. For GN50 (Figure 6A) those three peaks were still relatively well defined, as expected for 50 vol. % filled matrixes. Those peaks were almost unaffected by the addition of metallic nanoparticles, the main differences being: (i) the fusion peak was a bit less intense, (ii) the apparition of a diminutive transition peak around 117 °C, which might account for “core” LiBH_4_, filling mesopores, but not in close contact with the matrix [29,30]. This could be in line with our SEM observations that the metal might restrict LiBH_4_ from going deep within the matrix where there are smaller pores; still, it should not be related to “bulk” LiBH_4_ plainly outside of the matrix, as this species should present a sharp fusion peak at 275 °C [30]. For G2N50, the situation is a bit more appealing, as the peaks behaved distinctly. There the fusion peak was almost undistinguishable, and the peak of decomposition was lowered and flattened in the presence of Ni and Cu (while they were sharper and shifted toward higher temperatures for Co and Fe). Furthermore, the trace obtained with Ni did not display a peak of transition, which is a good indication of strong nanoconfinement effects. G2N50 Ni exhibited our lowest peak of decomposition (309 °C) at this filling value (Table 4).

#### 2.3.2. Volumetric Study

First hydrogen release

The volumetric studies are presented in Figure 7 to evaluate the functional behavior and reversibility of our materials. GN50 and G2N50 behaved particularly well, considering they are 50 vol. % filled materials, behaving even better than our previous N-matrixes filled at 30 vol. % (we recall that we augmented the proportion of graphene and N-dopant) [29]. Indeed, G2N50 liberated 1 wt. % H_2_ (versus m LiBH_4_) at 229 °C (15° lower than our previous best), and at 325 °C, it liberated 9.3 wt. % H_2_ (a 0.7 wt. % increase). Unfortunately, in most cases, the presence of metallic nanoparticles did not improve the functional properties of our hydrogen storage material (Table 5). At best, in the case of GN50 Fe, the presence of metallic nanoparticles did not affect the first and second cycle of hydrogen release, while for the same matrix in the presence of Co, Ni, and Cu nanoparticles to attain 1 wt. % of H_2_ released, the temperature had to be increased by 16 °C; for those same metals at 325 °C, the proportion of released H_2_ was reduced by 1 wt. %. In the case of G2N, the impact of metallic nanoparticles was surprisingly negative, reducing the onset temperature by 30 to 40 °C and lowering the H_2_ release at 325 °C by more than 2 wt. % in average. In any case, the hydrogen released per mass of LiBH_4_ was very close to its theoretical value (13.6 wt. %), which is in line with the oxides being reduced during the impregnation process as presented in Appendix A. If this was not the case, the available oxides were present in limited quantities (supposing all 5 wt. % Fe was oxidized to Fe_2_O_3_, in G2N at maximum 1 mmol of atomic O might be present per 5 mmol of LiBH_4_, reducing by 2.5 wt. % the liberated H_2_; this value decreases to 0.8 wt. % for GN Ni with 50% of oxidized Ni). Still, it cannot be ignored that a very slight decrease in hydrogen capacity was observed when the matrixes were covered by nanoparticles.

Second hydrogen release

Considering the second cycle, the impact of nanoparticles over hydrogen release was limited for GN, and a bit negative for G2N with a decrease of 0.6 wt. %. As in the previous study, matrixes decorated with Ni nanoparticles behaved better than other metals and inclusively better than the metal-free matrixes.

#### 2.3.3. Structural Impact of Material Cycling

The impact of two hydrogen release cycles over the material was evaluated by PXRD (Figure 8) and summarized in Table 6. No noticeable peaks were observed from cycled GN50 and G2N50, which is usual for LiBH_4_ degradation products in matrixes of reduced pore-size [20]. Noteworthily, the peak at 26° observed for G2N Fe and G2N Co was still present but far less intense. Wang et al. reported a decrease in the intensity of this peak under specific conditions [50]. Cu derivatives present very little change regarding their non-impregnated and just-impregnated relatives. In the case of Fe, Co, and Ni derivatives, cycling promoted the appearance of poorly resolved peaks attributed to the formation of Fe_2_B, CoB, and Ni_2_B species, as observed in our previous work [30]. According to Scherrer’s equation, those peaks correspond to crystals of limited size (<20 nm), except for the case of Ni_2_B (46 nm) and metallic Cu (100 nm). In the case of cGN50 Ni, a very weak peak was observed at 43.5°, attributed to a limited fraction of NiO.

## 3. Discussion

The decoration of GN matrixes by incipient wetness impregnation afforded Fe, Co, and Ni nanoparticles of 17 to 21 nm (20 to 30 nm on G2N matrixes) evenly distributed on the surface and throughout the thickness of the matrix, with no evidence of large metallic domains (unlike with manual grinding of the salt). Nevertheless, the same method afforded 150 nm Cu nanoparticles concentrated on the surface of the matrix. It illustrates that if a metal displays interesting theoretical features (as with Cu), its application in the real material might present difficulties not considered (such as aggregation) by the thermodynamics of reaction. Whereas incipient wetness impregnation displayed some advantages over manual grinding of the salt, it also promoted the formation of oxides in the cases of Fe, Co, and Ni. By impregnating the matrix with LiBH_4_ the particles appeared slightly larger and SEM observations indicated this could be due to the formation of a very thin LiBH_4_ film covering these. If it is the case, one should consider if the competition between matrix or nanoparticle wetting by LiBH_4_ might limit the performance of the system. Higher proportions of LiBH_4_ might also promote the clustering of nanoparticles.

The interaction between the nanoparticles and LiBH_4_ was confirmed by PXRD, with the formation of Fe_2_B, CoB and Ni_2_B. Interestingly, no metal borides were observed with Cu nanoparticles. Phase separation is a major limitation for the reversible degradation of LiBH_4_, as LiH and B segregate, particularly in carbon matrixes. It should be determined how metal borides and lithium hydride are locally distributed along several hydrogen release/capture cycles.

N-doped matrixes displayed interesting behavior, with notably low temperatures of hydrogen release. Raising the proportion of nitrogen and graphene dopants improved the material’s performance, suggesting that there is some room to improve our matrixes with nitrogen derivatives. On the other hand, while the matrix’s decoration with metallic nanoparticles was performed with good control over particle size and distribution, the impact on performance was negligible at best, antagonistic in some instances. It refutes our previous observations, where Ni, Co, and NiCo nanoparticles reproducibly improved N-doped matrixes [30]. Several factors should be highlighted to discuss these contradictions:Here, the proportion of metal was limited to 5 wt. %, while in our previous work, the proportion of nanoparticles was set to 5% of the mesopore volume, reaching a 27 wt. %;The matrixes’ pore size was lowered, and the proportion of N-dopant was increased, which are conditions more favourable to hydrogen release;Oxides were observed in the present work.

Factor 1 is crucial as, if a metal promotes hydrogen liberation, higher proportions of this metal are likely to display more impact, up to coalescence of the particles. Thus, metal catalysts usually present an optimum of activity at an intermediate weight percentage, where the particles are small enough to present a high surface/volume ratio, with enough mass to display noticeable activity. Nevertheless, higher proportions of dense metal will drastically reduce the hydrogen capacity of our materials. Considering the overall performance, it was crucial to determine if the activity of the added metallic dopant was catalytic or massive. The results showed that lowering a metal’s mass at the surface of the matrixes lessened activity, suggesting that catalytic quantities of metal are not enough to improve the material’s performance. The discrepancy between this work and the previous one [30] can highlight that a substantial amount of metal is required to observe an enhancement, thus posing the question of the method’s viability as mass capacity is the central parameter of this topic.

Regarding factor 2, in our previous work [30] the N-doped matrixes presented larger pores of broader distribution (9 nm), and its decoration with metallic nanoparticles lowered the onset of hydrogen liberation. Here we employed N-doped matrixes with well-defined pores of 4 to 6 nm, but their decoration with distinct nanoparticles was not beneficial for the performance of the material. This suggests that the effect of metallic nanoparticles does not synergize well with reducing the size of the pores, one effect taking dominance over the other. Moreover, if we compare this work to our previous N-doped matrixes, it appears that the results are quite comparable, even better in the present publication, probably because a higher proportion of nitrogen was doping the matrixes [29]. It is likely that hydrogen dissociation is a limiting factor for LiBH_4_ decomposition that can be tackled by nanoconfinement, nitrogen doping and/or metal catalyst. Nevertheless, if in matrixes of reduced pore size the hydrogen dissociation is not a limiting factor anymore, then the metal catalyst would lose its specific interest.

The relevance of factor 3 is more debatable as the oxides mostly disappeared after the impregnation process. Secondly, the presence of oxides, while detrimental to the reversibility of the hydrides, is not necessarily limiting their kinetics, as there is proof of faster hydrogen release in the presence of oxides [35,39,44,45,46]. To settle the origin of those oxides, we decided to decorate our matrix by manual grinding instead of incipient wetness impregnation. It appeared that by reproducing the manual grinding of the matrix with nickel nitrates we observed only the peaks of metallic nickel, as previously (Appendix A). This evidenced that the oxides originate from dissolving the nitrate in methanol. We are not sure how dissolving an aqueous nitrate in methanol can result in such a drastic change in our material after drying and pyrolysis, but we must assume that some coordination complexes were formed that hindered the complete reduction during the pyrolysis. Further calorimetric and volumetric characterizations of our solid-grinded material were performed, and very few differences were observed compared to the wet-impregnation protocol (Appendix A), discarding de facto the role of oxides over the poor performances of the nanoparticle’s decoration strategy.

Yet, if the lack of positive results observed for GN matrixes can be attributed to the insufficient quantities of metal and reveals the absence of synergies with the porous structure, we were surprised that, in the case of G2N, the metal-decorated materials performed even worse than their non-decorated counterparts. SEM observations and PXRD patterns illustrated the existence of an interaction between the metallic nanoparticles and just-impregnated LiBH_4_. In our ultimate work, we demonstrated that LiBH_4_ fills micropores first and these had much better performances than mesopores during the first cycle [31]. G2N matrixes present a huge proportion of micropores (43%) and our best performance so far. We were curious to see if the decline of the material’s performance could be related to LiBH_4_ wetting the matrix. In Appendix A we compare the wetting behavior of G2N with G2N Ni after impregnating it with LiBH_4_ at 50 vol. %. While non-impregnated G2N and G2N Ni display very similar textural characteristics, just-impregnated G2N50 displays many fewer empty micropores than its G2N50 Ni counterpart. It convincingly attests that the presence of nickel nanoparticles might hinder the complete wetting of the smaller pores of the matrix, as the metal is competing with the matrix for LiBH_4_^′^s contact. Interestingly, Gross et al. improved the wetting of their matrix by decoration with nickel nanoparticles, but what is noteworthy, they confined metallic magnesium instead of lithium borohydride, so the affinity of the impregnated material for the decorating particles differed widely [51]. It is also worth pointing out that, if decorated by Co or Fe, G2N displayed a decrease of the micropore volume (related to the appearance of graphitic domains at 2 ϴ = 26°) that could be related to their poor overall performance.

## 4. Materials and Methods

### 4.1. Synthesis of Materials

#### 4.1.1. Chemicals

All chemicals were employed as received, without purification. Graphite flakes were supplied by Aldrich (St. Louis, MO, USA, product number 33246-1) and hydrogen peroxide (30%), formaldehyde (40%), and ethylenediamine (98.0%) by Biopack (Buenos Aires, Argentina). Ascorbic acid (99.0%), potassium permanganate (99.0%), sulfuric acid (98%), hydrochloric acid (35%), diethyl ether (98.0%), ethanol (99.5%), and resorcinol (98.5%) were purchased from Cicarelli (San Lorenzo, Argentina) and phosphoric acid (85%) from Merck (Kenilworth, NJ, USA). Sodium carbonate decahydrate (99.99%) was provided by Timper. Co(NO_3_)_2_⋅6H_2_O was provided by Biopack (98.0%), while Ni(NO_3_)_2_⋅6H_2_O, Fe(NO_3_)_3_⋅9H_2_O, and Cu(NO_3_)_2_⋅XH_2_O were supplied by Sigma-Aldrich (97.0%). LiBH_4_ (Sigma-Aldrich, 90%) was milled prior to use. Every hydride material was handled within a glovebox to avoid air contact (content of oxygen and water < 5 ppm).

#### 4.1.2. Wet Synthesis

In brief, graphene oxide was prepared from graphite flakes according to the Tour method, employing potassium permanganate as oxidant in a mixture of sulfuric and phosphoric acid [52]. Later, the reaction mixture was poured at room temperature (RT) over ice with hydrogen peroxide to solubilize any manganese derivative. Then, it was washed, filtered, and centrifuged several times with Milli-Q water, diluted hydrochloric acid, ethanol, and finally coagulated with diethyl ether. After drying, a 6 mg/mL graphene oxide suspension in Milli-Q water was dispersed in an ultrasonic bath for 1 h and stored at 4 °C.

Each gel and following resin was prepared in a polypropylene bottle, screw-tapped before heating, to avoid any mechanical perturbation. N-doped hydrogels were formed by adding 10 mL of a recently sonicated (30 min) graphene oxide suspension (6 mg/mL) to a 1/5 diluted aqueous ethylenediamine solution (0.75 and 1.5 mmol for GN and G2N resp.) under magnetic stirring [53]. After a quick homogenization (1 min) the mixture was placed in an oven at 85 °C for 5 h. This temperature promoted gel shrinkage by 10%. The gel was washed several times with water at RT, then wet every 4 h with 3 g of the resorcinol−formaldehyde sol over 24 h, and the supernatant was discarded prior to the addition of the following aliquot. The sols were prepared with a mixture of water (5.43 g), resorcinol (13.05 g), formaldehyde (19.52 g), and sodium carbonate (71.9 mg). After the last washing, the gel was placed in an oven at 50 °C for 24 h and then at 90 °C for 72 h. Once cooled, the resins were washed with water and acetone and allowed to dry.

#### 4.1.3. Solid-State Synthesis

The resins were broken to smaller pieces (approx. 1 cm^3^ each, any red part without graphene was visually discarded) and pyrolyzed at 800 °C (3 °C/min from RT to 600 °C, 60 min dwelling at this temperature then heated to 800 °C at 3 °C/min and staying at this temperature for 6 h, then allowed to cool to RT at 3 °C/min over 3 h) to afford black chunks. Before impregnation, the resins and the LiBH_4_ were separately milled by employing a P6 Pulverisette planetary device, with an 80 cm3 milling chamber and five stainless balls, under an argon atmosphere. LiBH_4_ (1.5 g) were ball-milled for 300 min at 400 rpm, with a sequence of 10 min milling and 10 min pause. To reduce morphological impact and Fe contamination, the resins were milled for a shorter time: five reverse repetitions of 2 min milling at 200 rpm with a 1 min pause. The powders obtained from the resins were activated under reduced pressure (5 °C/min ramp then 3 h at 400 °C) prior to their introduction in the glove box.

To decorate the carbon material with metallic NPs, 10 mL of nitrate solutions (9.0 mM for Fe, 8.5 mM for Co, 8.5 mM for Ni, and 7.9 mM for Cu) were poured over 0.095 g of resin powder and agitated at RT (in the air) for 2 h. The amount of salt was calculated so that the mass of reduced metal would equal 5 wt. % of the matrix. The methanol was removed in a rotavapor at 65 °C until the powder looked dry on the glass. Then the powder was placed in a quartz tube that was degassed with N_2_ at 50 mL/min overnight. The mixture was heated at 800 °C (4 °C/min from RT to 200 °C, 2 h dwelling at this temperature then heated to 800 °C at 4 °C/min and staying at this temperature for 5 h, then allowed to cool to RT at 3 °C/min over 3 h) to promote the melting of the nitrate and its following reduction by the carbon surrounding material [54,55] These powders were activated at 400 °C under reduced pressure (5 °C/min ramp then 3 h at 400 °C) prior to their introduction in the glove box. To limit any oxidation of the metal nanoparticles, the resins were transferred quickly (<1 min) from the quartz tube of the furnace to the vacuum tube, which was opened after activation in the glove box.

The powders of resins (either decorated or not) and LiBH_4_ were manually mixed within the glovebox for 30 min using a mortar and pestle at 50 vol. %. The amount of LiBH_4_ employed to fill the resins at any given volume percentage was determined according to the volume of micropores + mesopores obtained from the nitrogen desorption isotherms.

#### 4.1.4. Melt Impregnation

The mixed powders of resins and LiBH_4_ were placed in a reactor within an autoclave and heated to 300 °C under 60 bar H_2_ for 30 min. At this temperature, the solid LiBH_4_ melts and the liquid LiBH_4_ wets the resin and fills its pores, while the high hydrogen pressure ensures that no hydrogen is liberated by the hydride. Once the system cooled back to RT, two options were taken: (i) the sample’s hydrogen capacity was directly evaluated using a modified Sieverts-type device and (ii) the impregnated sample was stored in the glove box to be submitted to other techniques.

### 4.2. Material Characterization

Textural parameters of the samples were studied using a Micromeritics ASAP 2020 analyzer. After surface cleaning in vacuum overnight, N2 adsorption/desorption isotherms were collected at −196 °C on 0.1 g of sample. For resins without LiBH_4_, this was done at a temperature of 300 °C, while if LiBH_4_ was present, the temperature was limited to 150 °C. The surface area and pore size distribution were obtained by the application of the Brunauer−Emmett−Teller (BET) and the Barrett−Joyner−Halenda (BJH) models, respectively. BET was determined for 0.03 < P/P_0_ < 0.12 with positive values of C. The mesopore volume was calculated according to BJH, and the total pore volume was determined by the Gurvich method at P/P_0_ = 0.96. The micropore volume and the external surface were estimated with the t-plot method. In this case, we used the standard reference t-curve for carbonaceous materials (carbon black) proposed by Magee [56].

Morphological and agglomerate size distribution analyses of the samples were performed by scanning electron microscopy (SEM, SEM-FIB, Zeiss, Crossbeam 340), the powders were dispersed over a carbon tape. Elemental analyses of the materials were also performed by energy-dispersive X-ray spectroscopy (EDXS) on SEM.

X-ray powder diffraction (PXRD) was realized with an air-tight chamber filled in a glove box, and the trace was recorded on a Bruker D8 ADVANCE apparatus using Cu Kα radiation.

The thermal desorption behavior of the hydride phases was studied by differential scanning calorimetry (DSC, TA q2000 calorimeter), using a heating rate of 5 °C/min and an argon flow rate of 122 mL/min. The samples were placed in a closed aluminum holder within the glovebox to minimize air contact. For each plot, the heat flow was normalized with respect to the mass of LiBH_4_.

Hydrogen sorption kinetic measurements were obtained using modified Sieverts-type equipment coupled with a mass flow controller. The sample was placed in a stainless reactor, within an autoclave that was connected to the Sieverts device. Dehydrogenation curves were obtained by heating up to 400 °C with a hydrogen back pressure of 0.5 ± 0.1 bar. The amount of absorbed/desorbed hydrogen is expressed as the H_2_/LiBH_4_ mass ratio and is determined with a relative error of ± 5%. Rehydrogenation was performed at 400 °C with a sudden increase of the pressure to 60 bar and keeping the sample overnight and measuring the evolution of pressure (typically higher than 50 bar). After cooling, a second dehydrogenation was performed and the sample was collected for further characterization (noted “c”)

In the ensuing discussion, the samples are presented accordingly to the following codes: non-decorated (GN and G2N, black), decorated with Fe (orange), Co (red), Ni (green), Cu (blue). The samples just impregnated with LiBH_4_ are noted by a number indicating the volumetric percentage of LiBH_4_ filling the matrix (for example GN50). The cycled samples (submitted to dehydrogenation/rehydrogenation/dehydrogenation) are highlighted by a “c” prefix (for example cGN50).

## 5. Conclusions

Two N-doped matrixes presenting micropores and small mesopores (3.8 and 6.1 nm) were decorated at 5 wt. % by incipient wetness impregnation using Fe, Co, Ni, and Cu nitrates in methanol. Fe-, Co-, and Ni-decorated matrixes presented particles of limited size (20 nm) homogeneously distributed over and within the matrix, while Cu nanoparticles were larger (160 nm) and concentrated on the surface. Only metallic Cu was observed by PXRD, while the matrixes decorated with Fe, Co, and Ni also presented oxides, attributed to the incipient wetness impregnation. The oxides disappeared after LiBH_4_ was impregnated at 50 vol. % and metal borides were observed after two hydrogen/release cycles. According to DSC studies, the impact of metal nanoparticles on the calorimetric behavior of the material was limited. Volumetric experiments revealed that the presence of metallic nanoparticles did not improve the hydrogen release properties of GN matrixes and induced even worse behavior for G2N. We proposed that the low proportion of metal and their lack of synergies with pores of reduced size might make it harder to improve the already good behavior of N-doped matrixes. In the case of micropore-rich G2N, the competition between LiBH_4_ pore-wetting and LiBH_4_ metal-wetting could be responsible for the observed decrease in the material’s performance.

## Figures and Tables

**Figure 1 molecules-27-02921-f001:**
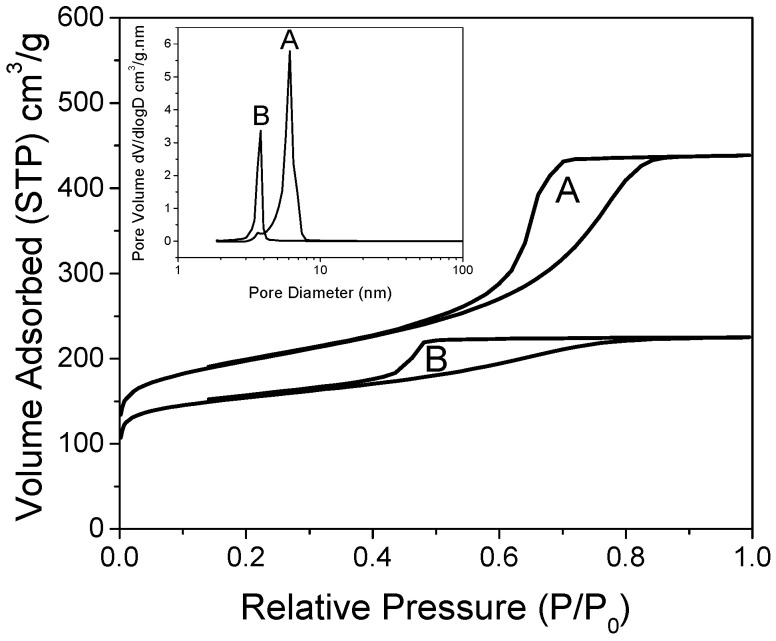
Nitrogen isotherms of GN (A) and G2N (B), with corresponding pore-size distribution obtained by BJH (inset).

**Figure 2 molecules-27-02921-f002:**
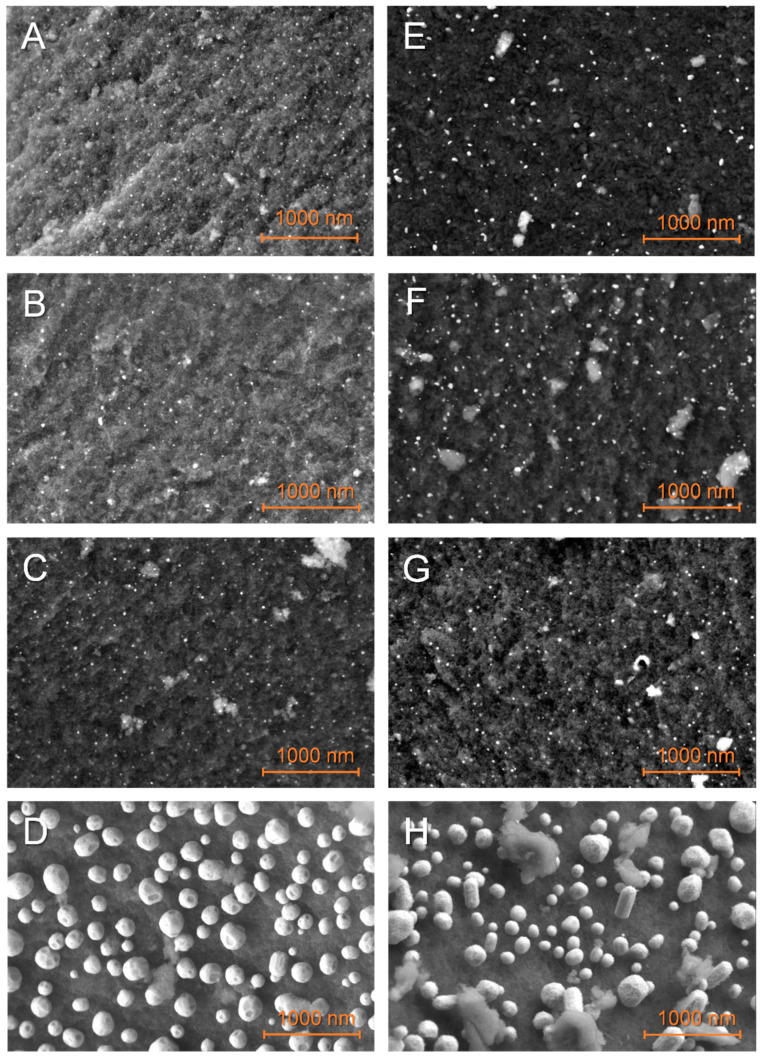
SEM observations of the non-impregnated GN (**A**–**D**) and G2N (**E**–**H**) matrixes decorated with Fe (**A**,**E**), Co (**B**,**F**), Ni (**C**,**G**), Cu (**D**,**H**) nanoparticles.

**Figure 3 molecules-27-02921-f003:**
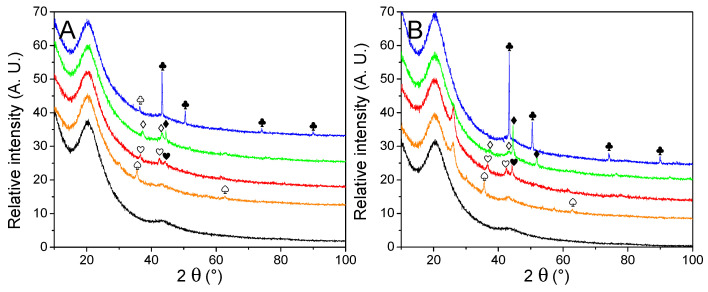
PXRD patterns of non-impregnated GN (**A**) and G2N (**B**) matrixes (black) decorated with Fe (orange), Co (red), Ni (green) and Cu (blue) nanoparticles. Crystalline structures are marked for Fe (♠), Co (♥), Ni (♦), Cu (♣). Respective oxides are indicated with empty symbols (♤♡♢♧). A possible graphite species is marked for G2N Fe and G2N Co ( ).

**Figure 4 molecules-27-02921-f004:**
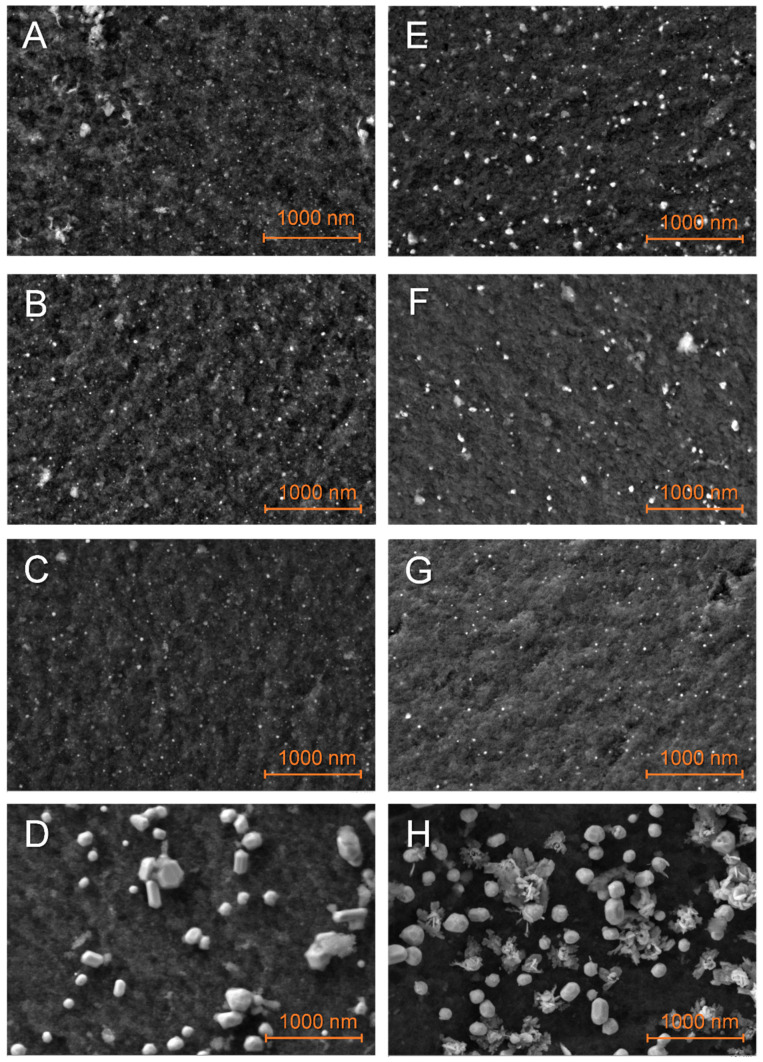
SEM observations of the just-impregnated GN50 (**A**–**D**) and G2N50 (**E**–**H**) matrixes decorated with Fe (**A**,**E**), Co (**B**,**F**), Ni (**C**,**G**), Cu (**D**,**H**) nanoparticles.

**Figure 5 molecules-27-02921-f005:**
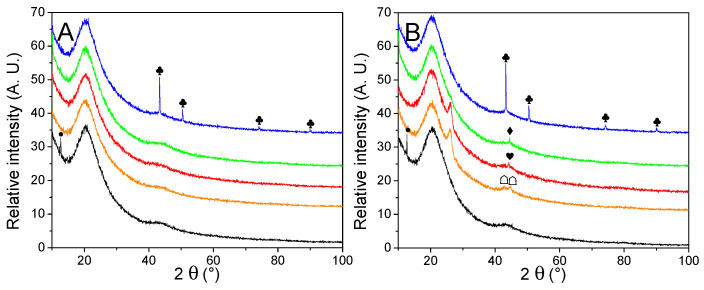
PXRD traces of just-impregnated GN50 (**A**) and G2N50 (**B**) matrixes (black) decorated with Fe (orange), Co (red), Ni (green) and Cu (blue) nanoparticles. Crystalline structures are marked for Co (♥), Ni (♦), Cu (♣). Impregnated LiBH_4_ is marked in GN50 and G2N50, assuming crystal order loss (∙). Fe_2_B is marked (☖). A possible graphite species is marked for G2N Fe and G2N Co ( ).

**Figure 6 molecules-27-02921-f006:**
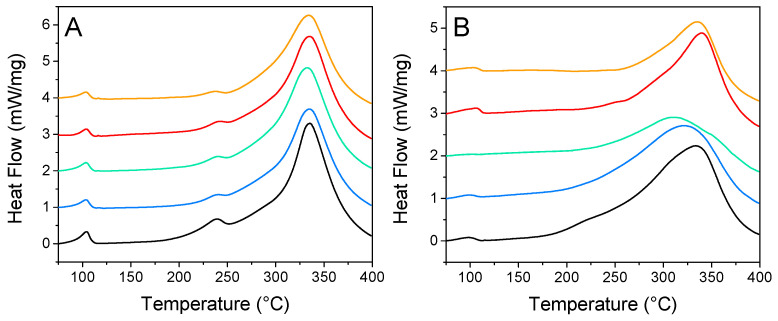
Differential scanning calorimetry from GN (**A**) and G2N (**B**) filled with LiBH_4_ at 50% (vol) without (black) and with Fe (orange), Co (red), Ni (green), and Cu (blue) nanoparticles.

**Figure 7 molecules-27-02921-f007:**
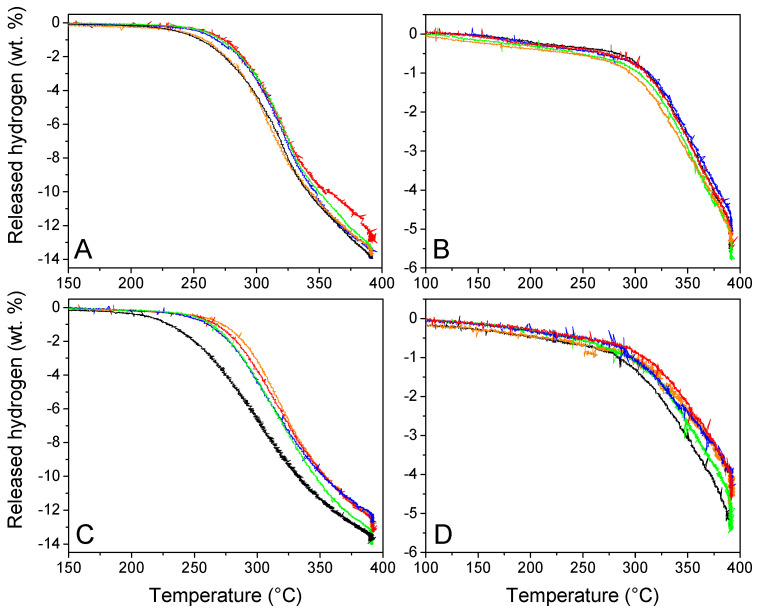
Volumetric studies of GN50 (**A**,**C**) G2N50 (**B**,**D**) without (black) and with Fe (orange), Co (red), Ni (green), and Cu (blue) nanoparticles during the first (**A**,**C**) and second (**B**,**D**) dehydrogena-tion cycles.

**Figure 8 molecules-27-02921-f008:**
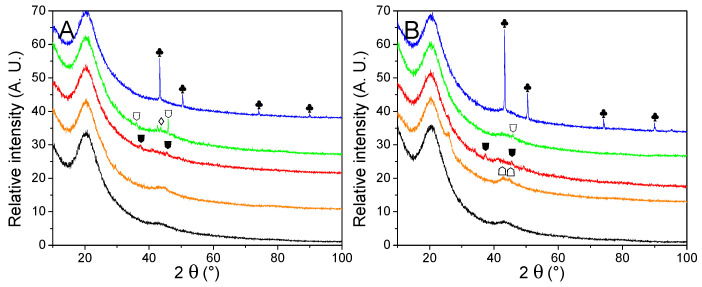
PXRD traces of cycled GN50 (**A**) and G2N50 (**B**) matrixes (black) decorated with Fe (orange), Co (red), Ni (green) and Cu (blue) nanoparticles. Crystalline structures are marked for Cu (♣). Nickel oxides are indicated (♢). Metal borides are noted for Fe_2_B (☖), CoB (⛊) and Ni_2_B (⛉). A possible graphite species is marked for G2N Fe and G2N Co ( ).

**Table 1 molecules-27-02921-t001:** Textural parameters of matrixes GN and G2N with and without decoration by metallic nanoparticles before their impregnation by LiBH_4_.

Matrix Type	Metallic NP	S_BET_ ^a^ (m^2^/g)	S_ext_ ^b^ (m^2^/g)	S_micro_ ^c^ (m^2^/g)	V_tot_ ^d^ (cm^3^/g)	V_meso_ ^e^ (cm^3^/g)	V_micro_ ^c^ (cm^3^/g)	D_max_ ^e^ (nm)
GN	None	720	330	390	0.68	0.48	0.16	6.1
	Fe	670	300	360	0.63	0.46	0.15	6.0
	Co	610	280	330	0.60	0.45	0.14	6.1
	Ni	660	300	370	0.63	0.45	0.15	6.1
	Cu	690	300	390	0.65	0.42	0.16	6.1
	NP *	660 ± 30	290 ± 10	360 ± 20	0.63 ± 0.02	0.45 ± 0.02	0.15 ± 0.01	6.1
G2N	None	580	190	390	0.35	0.17	0.16	3.8
	Fe	360	210	150	0.31	0.23	0.06	3.8
	Co	360	180	180	0.31	0.22	0.08	3.8
	Ni	520	180	340	0.32	0.17	0.14	3.8
	Cu	500	170	330	0.31	0.16	0.13	3.8
	NP *	430 ± 90	180 ± 20	250 ± 90	0.31 ± 0.01	0.20 ± 0.03	0.10 ± 0.04	3.8

Values determined by ^a^ BET, ^b^ BET-t-plot, ^c^ t-plot, ^d^ Gurvich, ^e^ BJH. * Average and standard deviation of the parameter obtained from all four metal-decorated matrixes.

**Table 2 molecules-27-02921-t002:** Crystallographic parameters of the NPs obtained by XRPD of the free matrixes.

Matrix Type	Metallic NP	Average Size ^a^ (nm)	Dc ^b^ (nm)	Peak Position (°) ^c^	Attributed Species
GN	Fe	17 ± 9	19	**35.6**; 62.6	Fe_2_O_3_/Fe_3_O_4_
	Co	21 ± 9	13 11	44.2 36.5; **42.5**	Co CoO
	Ni	20 ± 7	33 11	44.5 37.1; **43.3**	Ni NiO
	Cu	160 ± 60	103 50	**43.3**; 50.4; 74.1 36.4	Cu CuO
G2N	Fe	20 ± 10	23	35.4; 62.5	Fe_2_O_3_/Fe_3_O_4_
	Co	30 ± 20	29 11	44.2 36.5; **42.5**	Co CoO
	Ni	21 ± 7	37 8	**44.5**; 51.9 43.3	Ni NiO
	Cu	170 ± 50	105	**43.3**; 50.4; 74.1	Cu

^a^ Average particle size determined by manual counting on SEM images, ^b^ average crystallite size determined with the peak of highest intensity according to Scherrer’s formula, ^c^ peak of highest intensity of a given species is indicated in bold.

**Table 3 molecules-27-02921-t003:** Crystallographic parameters of the NPs obtained by PXRD of matrixes just impregnated with LiBH_4_.

Matrix Type	Metallic NP	Average Size ^a^ (nm)	Dc ^b^ (nm)	Peak Position (°) ^c^	Attributed Species
GN50	Fe	18 ± 6	-	-	-
	Co	17 ± 7	-	-	-
	Ni	24 ± 9	-	-	-
	Cu	160 ± 90	98	**43.3**; 50.4	Cu
G2N50	Fe	20 ± 20	10	42.5; **45.0**	Fe_2_B
	Co	30 ± 10	33	44.4	Co
	Ni	22 ± 6	37	44.5	Ni
	Cu	170 ± 80	89	**43.3**; 50.4; 74.1	Cu

^a^ Average particle size determined by manual counting on SEM images, ^b^ average crystallite size determined with the peak of highest intensity according to Scherrer’s formula, ^c^ peak of highest intensity of a given species is indicated in bold.

**Table 4 molecules-27-02921-t004:** Calorimetric behavior of LiBH_4_-impregnated GN and G2N matrixes with and without metallic decoration.

Matrix Type	Metallic NP	T_t_ ^a^	T_f_ ^b^	T_d_ ^c^
GN50	None	104	239	335
	Fe	103 (116)	238	334
	Co	104 (116)	242	335
	Ni	103 (116)	240	332
	Cu	103 (117)	240	335
G2N50	None	98 (115)	219	333
	Fe	103	-	335
	Co	105	252	339
	Ni	99	-	309
	Cu	99	-	320

^a^ Temperature of orthorhombic to hexagonal transition (°C). ^b^ Melting temperature (°C). ^c^ Temperature of decomposition of the hydride (°C).

**Table 5 molecules-27-02921-t005:** Hydrogen release capacity of GN50 and G2N50 with and without decoration.

Matrix type	Metallic NP	T_1%_ ^a^	Wt._325_ % ^b^	Wt.^rev^ % ^c^
GN50	None	255	8.0	5.6
	Fe	258	8.3	5.3
	Co	274	6.8	5.5
	Ni	271	6.9	5.8
	Cu	268	7.3	5.1
G2N50	None	229	9.3	5.4
	Fe	272	6.8	4.6
	Co	268	7.0	4.6
	Ni	263	7.6	5.5
	Cu	260	7.4	4.4

^a^ Temperature (°C) to attain 1 wt. % of liberated H_2_. ^b^ Amount of liberated H_2_ liberated at 325 °C with respect to infiltrated LiBH_4_. ^c^ Maximum value of H_2_ released at 400 °C during second dehydrogenation.

**Table 6 molecules-27-02921-t006:** Crystallographic parameters of the NPs obtained by PXRD of the cycled matrixes.

Matrix Type	Metallic NP	Dc ^a^ (nm)	Peak Position (°) ^b^	Attributed Species
cGN50	Fe			
	Co	4	37.3; **45.6**	CoB
	Ni	46 21	36.1; 42.7; **46.0** 43.5	Ni_2_B NiO
	Cu	112	**43.3**; 50.4; 74.1	Cu
cG2N50	Fe	14	42.5; **45.0**	Fe_2_B
	Co	21	37.3; **45.6**; 41.1; 49.2	CoB
	Ni	16	46.0	Ni_2_B
	Cu	98	**43.3**; 50.4; 74.1; 89.9	Cu

^a^ average crystallite size determined with the peak of highest intensity according to Scherrer’s formula, ^b^ peak of highest intensity of a given species is indicated in bold.

## Data Availability

Not applicable.

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
