# Peer review of "From Iron to Copper: The Effect of Transition Metal Catalysts on the Hydrogen Storage Properties of Nanoconfined LiBH4 in a Graphene-Rich N-Doped Matrix"

_molecules, 2022, doi:10.3390/molecules27092921_

Round 1

Reviewer 1 Report

Dear Editor

The authors employed the incipient-wetness impregnation to decorate N-doped graphene-rich matrixes with iron, nickel, cobalt, and copper nanoparticles. The quantities of metallic nanoparticles catalyse was determined by the mass of the metal fraction. This study is interesting and can be published in this journal after some minor revisions.

  1. The nanoparticles in Figure 4 D seem larger than these in Figure 4 H, can you give more expiations about it?
  2. The annotation for different color curves should be added in Figure 5, Figure 6, Figure 7 and Figure 8.
  3. The authors may consider citing more papers for a broader readership. For example,doi:10.1039/c7nr02502e;10.1021/acsami.6b15862. 

Author Response

  1. Histograms are more reliable than a single image and were presented in SI7. Indeed, particles are a bit bigger for G2N50 Cu (observed in figure 4 H) than for GN50 Cu (observed in figure 4 D), in accordance with the results exposed in table 3. It was globally observed that nanoparticles are a bit bigger on G2N than on GN, and we attributed this to the increased interaction of the matrix with the metal, which limits the formation of well-defined nanoparticles.

The general observation was already stated on lines 178-185: « The manual measure of at least 300 nanoparticles from five or more representative pictures afforded their statistical size, presented in Table 2 and distribution histograms (Figure SI 2). These data and the careful observation of higher-magnification images (Figure SI 3) indicate that the materials with more nitrogen (G2N) present bigger particles with broader distribution and a less defined shape. By comparing figures 2D and 2H, the latter present particles of less defined outer shape, suggesting that the nitrogen-metal interaction is higher and restrains the formation of Cu crystals (see also Figure SI 3 D, H and Figure SI 4). » 

Regarding the specific impregnated sample it was stated on lines 264-267: « Particles on matrixes just impregnated with LiBH4 appear slightly larger (Figure 4), with a broader distribution (Figure SI 7), denoting an interaction between the metal and the hydride. This interaction is particularly impressive in the case of G2N Cu (Figure 4H), with splatter-like nanoparticles (see elemental attribution in Figure SI Cu 3). »

To clarify, a sentence was added at the end of the paragraph: « Similar to the non-impregnated samples, bigger and less-defined nanoparticles are observed for higher proportions of N-dopant. »

  1. We checked those figures, but we believe the captions specify the color attribution already:

Figure 5. PXRD traces of just-impregnated GN50 (A) and G2N50 (B) matrixes (black) decorated with Fe (orange), Co (red), Ni (green) and Cu (blue) nanoparticles. Crystalline structures are marked for Co (♥), Ni (♦), Cu (♣). Impregnated LiBH4 is marked in GN50 and G2N50, assuming crystal order loss (·). Fe2B is marked (☖). A possible graphite species is marked for G2N Fe and G2N Co (ï‚·).

Figure 6. Differential scanning calorimetry from GN (A) and G2N (B) filled with LiBH4 at 50 % (vol) without (black) and with Fe (orange), Co (red), Ni (green), and Cu (blue) nanoparticles.

Figure 7. Volumetric studies of GN50 (A, C) G2N50 (B, D) without (black) and with Fe (orange), Co (red), Ni (green), and Cu (blue) nanoparticles during the first (A, C) and second (B, D) dehydrogenation cycles.

Figure 8. PXRD traces of cycled GN50 (A) and G2N50 (B) matrixes (black) decorated with Fe (orange), Co (red), Ni (green) and Cu (blue) nanoparticles. Crystalline structures are marked for Cu (♣). Nickel oxides are indicated (♢). Metal borides are noted for Fe2B (☖), CoB (⛊) and Ni2B (⛉). A possible graphite species is marked for G2N Fe and G2N Co (ï‚·).

  1. Initially, we thought this article should focus very specifically on the topic of nanoconfined LiBH4. Yet, as all reviewers asked us to improve and generalize the introduction, we understood we must change this point. Thus, several general audience cites were added, specifically on graphene and graphene application. Yet, we could not implement the two articles that the reviewer stated, as we thought graphene field-effect was too distant from the topic of this work, even if we understand the relation with metal-graphene effect

ADDED references:

  1. Abe, J. O.; Popoola, A. P. I.; Ajenifuja, E.; Popoola, O. M. Hydrogen energy, economy and storage: Review and recommendation. J. Hydrogen Energy 2019, 44, 15072-15086.
  2. Rusman, N. A. A., Dahari, M. A review on the current progress of metal hydrides material for solid-state hydrogen storage applications. J. Hydrogen Energy 2016, 41, 12108-12126.
  3. Daulbayev, C.; Lesbayev, B.; Bakbolat,B.; Kaidar ,B.; Sultanov ,F.; Yeleuov ,M.; Ustayeva ,G.; Rakhymzhan, N. A mini-review on recent trends in prospective use of porous 1D nanomaterials for hydrogen storage. Afr. J. Chem. Eng. 2022, 39, 52-61.
  4. Kumar, P.; Singh, S.; Hashmi, S.A.R.; Kim, K.-H. MXenes: Emerging 2D materials for hydrogen storage. Nano Energy 2021, 85, 105989
  5. Shet, S. P.; Shanmuga Priya, S.; Sudhakar, K.; Tahir, M. A review on current trends in potential use of metal-organic framework for hydrogen storage. J. Hydrogen Energy 2021, 46, 11782-11803.
  6. Rueda, M.; Sanz, L. M.; Martín, A. Innovative methods to enhance the properties of solid hydrogen storage materials based on hydrides through nanoconfinement: a review. Supercritic. Fluids 2018, 141, 198-217.
  7. Le, T. T.; Pistidda, C.; Nguyen, V. H.; Singh, P.; Raizada, P.; Klassen, T.; Dornheim, M. Nanoconfinement effects on hydrogen storage properties of MgH2 and LiBH4. J. Hydrogen Energy 2021, 46, 23723-23736.
  8. Verón, M. G., Troiani, H., Gennari, F. C. Synergetic effect of Co and carbon nanotubes on MgH2 sorption properties. Carbon 2011, 49, 2413–2423.
  9. Parambhath, V. B.; Nagar, R.; Ramaprabhu, S. Effect of Nitrogen Doping on Hydrogen Storage Capacity of Palladium Decorated Graphene. Langmuir 2012, 28, 7826-7833.
  10. Ngene, P., Van Zwienen, R., De Jongh, P.E. Reversibility of the hydrogen desorption from LiBH4: A synergetic effect of nanoconfinement and Ni addition. Commun. 2010, 46, 8201-8203.
  11. Xian, K.; Nie, B.; Li, Z.; Gao, M.; Li, Z.; Shang, C.; Liu, Y.; Guo, Z.; Pan, H. TiO2 decorated porous carbonaceous network structures offer confinement, catalysis and thermal conductivity for effective hydrogen storage of LiBH4. Eng. J. 2021, 407, 127156.
  12. Wang, S.; Gao, M.; Xian, K.; Li, Z.; Shen, Y.; Yao, Z.; Liu, Y.; Pan, H. LiBH4 Nanoconfined in Porous Hollow Carbon Nanospheres with High Loading, Low Dehydrogenation Temperature, Superior Kinetics, and Favorable Reversibility. ACS Appl. Energy Mater. 2020, 3, 3928−3938.
  13. Chen, X.; Li, Z.; Zhang, Y.; Liu, D.; Wang, C.; Li, Y.; Si, T.; Zhang, Q. Enhanced Low-Temperature Hydrogen Storage in Nanoporous Ni-Based Alloy Supported LiBH4. Chem. 2020, 8, 283.
  14. SI, A.; Kyzas, G. Z.; Pal, K.; de Souza, F. G. Graphene functionalized hybrid nanomaterials for industrial-scale applications: A systematic review. Mol. Struct. 2021, 1239, 130518.

15. Gross, A. F.; Ahn, C. C.; Van Atta, S. L.; Liu, P.; Vajo, J. J. Fabrication and hydrogen sorption behaviour of nanoparticulate MgH2 incorporated in a porous carbon host. Nanotechnology 2009, 20, 204005.

Reviewer 2 Report

Martinez et.al., studied the effect of transition metal dopants on the H2 storage properties of LiBH4 in N-doped graphene matrix. Several structural characterization studies were undertaken to understand the synergy between transition metal dopant and pore size distribution. Finally, H2 storage and release capacity of the doped graphene matrices were evaluated and it was concluded that none of Fe, CO, Ni and Cu are able to promote the H2 storage properties. 

This is a very extensive study. The conclusions are well supported by the experimental findings. However, I am little concerned that none of the dopants were able to promote the performance of the N-doped graphene matrices. Although the study seem complete, because of the negative results (unsuccessful promotion effects of the transition metals) I will leave the final decision to the editor. 

If the authors can show one dopant with promotes the H2 storage properties, I would be happy to accept the manuscript with additional findings.

Author Response

We understand the final decision is dependent on the orientation of the editor according to the concern formulated rightly by the reviewer.

Thus please let us present our arguments:

  • The current article is already presenting a lot of data and we believe more would be too much. Extensive work must be done for each metal, adding a single metal is usually enough work for a separate article. We are already considering other metals for future publications
  • The list of metals was not randomly selected, we were interested to assess especially the metal presented by Huang et al. (cite 41) and easily accessible as nitrates to compare the result of simulated bulk LiBH4 to experimental nanoconfined LiBH4
  • It is commonly stated in review articles that mixing strategies (nanoconfinement, catalyst, doping) must be considered to improve further the performance of hydrides. Yet, this strategy is not broadly presented, and we believe it is good also to show that not everything is as straightforward as it may seem
  • We compared to an already very efficient matrix, lowering metal quantities

Finally, despite we do not present positive results, it motivated us to show a lot of complementary data that explain convincingly why adding a metal catalyst might not improve the performance of an already kinetically accelerated material and can even hinder its performance. We believe this discussion might interest a lot of the readership, particularly in a special edition

Reviewer 3 Report

The manuscript entitled "FROM IRON TO COPPER: THE EFFECT OF TRANSITION METAL CATALYSTS ON THE HYDROGEN STORAGE PROPERTIES OF NANOCONFINED LIBH4 IN A GRAPHENE-RICH N-DOPED MATRIX" focuses on the influence of limited quantities of coordination metals on the hydrogen storage properties of nanoconfined LiBH4 and critical parameters that might restrain the synergies between nanoconfinement and the presence of metal catalysts. The comments may be useful for the improvement of the manuscript. Minor revisions are needed to make the work acceptable. 

  1. XPS analysis should be used to estimate the nitrogen content of the products and identify the functional groups for a better understanding the distribution of nitrogen among the structure.
  2. The authors tried to conclude that "The decoration of GN matrixes by incipient wetness impregnation afforded Fe, Co, 396 and Ni nanoparticles of 17 to 21 nm (20 to 30 nm on G2N matrixes) evenly distributed on 397 the surface and through the thickness of the matrix, with no evidence of large metallic 398 domains (unlike to manual grinding of the salt). Nevertheless, the same method afforded 399 150 nm Cu nanoparticles concentrated on the surface of the matrix". But the reviewer does not find the relevant results to demonstrate that.
  3. The introduction and discussion parts can be improved by providing a more critical discussion of recent related literature for example, some papers:

https://doi.org/10.1016/j.sajce.2021.11.008

https://doi.org/10.1021/la301232r

https://doi.org/10.1016/j.matlet.2020.127919

https://doi.org/10.1039/D1DT01498F

https://doi.org/10.1016/j.apsusc.2021.149176

Author Response

  1. XPS was already performed on this kind of matrixes and was presented in our previous article (cite 29). A sentence was added to clarify this in the current article on lines 123-124: “X-ray photoelectron spectroscopy (XPS) of these materials can be found in our previous work.”
  2. This affirmation was justified on lines 190-201: “Elemental analysis by EDS (Energy Dispersive Microscopy) of the surface of our materials at distinct energies (3 KeV, 15 KeV) indicated that Fe, Co, or Ni metals were present at val-ues very close to the expected 5 wt. % (see the corresponding SI M Map files for the EDS). Given the penetration of the beam is highly dependent on its energy (50 nm at 3 keV, 1 um at 15 keV), and as similar results were obtained at distinct energies, the repartition of the metal is likely homogeneous through the thickness of the matrix. It suggests that incipient wetness impregnation efficiently distributes those metals within the whole sample, at least up to the penetrating value of the beam of higher energy. On the other hand, for Cu-decorated matrixes, EDS at 3 keV revealed a mass concentration closer to 20 wt. %, while at 15 keV this value dropped to 5 wt. %. As Cu nanoparticles tend to agglomerate to form much larger structures, they are likely less able to remain within a constricted pore.”

EDS are presented several times for each metal. In the case of Fe, Co, Ni a close to 5 wt. % was observed independently to the energy of the beam. Firstly, at low energy, if the particles were concentrated on the surface, a higher count should be obtained. Secondly, at higher energy, the consistence of the count strongly indicates that the particles are as present at the surface than at the core of the matrix. It is not the case for Cu and is stated.

We consider the remark of the reviewer means a clarification was needed from our article, but we did not find a clearer way to explain it.

  1. We though this article should focus specifically on the topic of nanoconfined LiBH4. Yet, as all reviewer asked us to improve and generalize the introduction, we understood we must change this point. Thus, the introduction was extended to implement several general audience cites, some suggested by this reviewer

ADDED references:

  1. Abe, J. O.; Popoola, A. P. I.; Ajenifuja, E.; Popoola, O. M. Hydrogen energy, economy and storage: Review and recommendation. J. Hydrogen Energy 2019, 44, 15072-15086.
  2. Rusman, N. A. A., Dahari, M. A review on the current progress of metal hydrides material for solid-state hydrogen storage applications. J. Hydrogen Energy 2016, 41, 12108-12126.
  3. Daulbayev, C.; Lesbayev, B.; Bakbolat,B.; Kaidar ,B.; Sultanov ,F.; Yeleuov ,M.; Ustayeva ,G.; Rakhymzhan, N. A mini-review on recent trends in prospective use of porous 1D nanomaterials for hydrogen storage. Afr. J. Chem. Eng. 2022, 39, 52-61.
  4. Kumar, P.; Singh, S.; Hashmi, S.A.R.; Kim, K.-H. MXenes: Emerging 2D materials for hydrogen storage. Nano Energy 2021, 85, 105989
  5. Shet, S. P.; Shanmuga Priya, S.; Sudhakar, K.; Tahir, M. A review on current trends in potential use of metal-organic framework for hydrogen storage. J. Hydrogen Energy 2021, 46, 11782-11803.
  6. Rueda, M.; Sanz, L. M.; Martín, A. Innovative methods to enhance the properties of solid hydrogen storage materials based on hydrides through nanoconfinement: a review. Supercritic. Fluids 2018, 141, 198-217.
  7. Le, T. T.; Pistidda, C.; Nguyen, V. H.; Singh, P.; Raizada, P.; Klassen, T.; Dornheim, M. Nanoconfinement effects on hydrogen storage properties of MgH2 and LiBH4. J. Hydrogen Energy 2021, 46, 23723-23736.
  8. Verón, M. G., Troiani, H., Gennari, F. C. Synergetic effect of Co and carbon nanotubes on MgH2 sorption properties. Carbon 2011, 49, 2413–2423.
  9. Parambhath, V. B.; Nagar, R.; Ramaprabhu, S. Effect of Nitrogen Doping on Hydrogen Storage Capacity of Palladium Decorated Graphene. Langmuir 2012, 28, 7826-7833.
  10. Ngene, P., Van Zwienen, R., De Jongh, P.E. Reversibility of the hydrogen desorption from LiBH4: A synergetic effect of nanoconfinement and Ni addition. Commun. 2010, 46, 8201-8203.
  11. Xian, K.; Nie, B.; Li, Z.; Gao, M.; Li, Z.; Shang, C.; Liu, Y.; Guo, Z.; Pan, H. TiO2 decorated porous carbonaceous network structures offer confinement, catalysis and thermal conductivity for effective hydrogen storage of LiBH4. Eng. J. 2021, 407, 127156.
  12. Wang, S.; Gao, M.; Xian, K.; Li, Z.; Shen, Y.; Yao, Z.; Liu, Y.; Pan, H. LiBH4 Nanoconfined in Porous Hollow Carbon Nanospheres with High Loading, Low Dehydrogenation Temperature, Superior Kinetics, and Favorable Reversibility. ACS Appl. Energy Mater. 2020, 3, 3928−3938.
  13. Chen, X.; Li, Z.; Zhang, Y.; Liu, D.; Wang, C.; Li, Y.; Si, T.; Zhang, Q. Enhanced Low-Temperature Hydrogen Storage in Nanoporous Ni-Based Alloy Supported LiBH4. Chem. 2020, 8, 283.
  14. SI, A.; Kyzas, G. Z.; Pal, K.; de Souza, F. G. Graphene functionalized hybrid nanomaterials for industrial-scale applications: A systematic review. Mol. Struct. 2021, 1239, 130518.
  15. Gross, A. F.; Ahn, C. C.; Van Atta, S. L.; Liu, P.; Vajo, J. J. Fabrication and hydrogen sorption behaviour of nanoparticulate MgH2 incorporated in a porous carbon host. Nanotechnology 2009, 20, 204005.

Round 2

Reviewer 2 Report

I agree with authors' response. 

I am okay with accepting this article for publication.